

# Gene expression profiling of extraocular muscles in primary inferior oblique overaction

Rui Hao[1,2,3,4,*], Yuchuan Wang[1,2,3,4,*] and Wei Zhang[1,2,3,4]

[1] Tianjin Eye Hospital, Tianjin, China
[2] Tianjin Key Laboratory of Ophthalmology and Visual Science, Tianjin, China
[3] Nankai University Affiliated Eye Hospital, Tianjin, China
[4] Clinical College of Ophthalmology, Tianjin Medical University, Tianjin, China
[*] These authors contributed equally to this work.

Corresponding authors
Rui Hao, haorui0311@126.com
Wei Zhang, zhangwei_eye@163.com

## ABSTRACT

**Background**. This study investigates gene expression differences in primary inferior oblique overaction (IOOA) by performing transcriptome sequencing on extraocular muscles (EOMs) from patients with primary and secondary IOOA. Strabismus, particularly IOOA, is often associated with abnormal eye movement due to imbalanced muscle function. By using bioinformatic analyses to identify differentially expressed genes (DEGs) and enriched pathways, we aim to uncover the molecular distinctions that may underlie the unique neuromuscular characteristics of primary IOOA.

**Methods**. Transcriptome sequencing was conducted on EOMs from ten patients with primary IOOA and ten patients with secondary IOOA. DEGs were identified using DESeq2. Gene Ontology (GO) annotations were enriched using ClusterProfiler and GlueGo, and the overall gene expression data were analyzed with Gene Set Enrichment Analysis (GSEA). The protein-protein interaction (PPI) network of DEGs was constructed using STRING.

**Results**. We identified 258 DEGs, with 110 genes significantly upregulated and 148 genes downregulated in the primary IOOA group compared to the secondary IOOA group. Analysis of DEGs revealed that upregulated genes in the primary group were associated with myelination (*e.g.*, MBP, MPZ, PRX) and ion channels (*e.g.*, KCNA5, KCNE5). Conversely, downregulated genes were primarily related to ion channels (*e.g.*, CACNA1B, SCN3B, SCN5A, KCNJ3), collagen fibril organization (*e.g.*, COL11A1, COL11A2, COL22A1, COL25A1 and COL9A2). Further analysis of cellular components and molecular functions indicated that genes related to M-bands (*e.g.*, MYOM2, MYOM3) were upregulated in the primary group. GSEA and PPI network analysis corroborated these findings, highlighting alterations in peripheral nervous system development and myelin sheath formation.

**Conclusion**. Our preliminary findings suggested that neuromuscular DEGs in primary IOOA were enriched in pathways related to myelination, ion channels, M-bands, and collagen fibril. The results indicated that multiply innervated muscle fibers were more abundant in the primary IOOA group, likely enabling slow tension generation, whereas the secondary IOOA group exhibited higher collagen fibril levels that might improve muscle stiffness.

## INTRODUCTION

The diversity of muscle fiber types is influenced by the activation of genes specific to different muscles, including those of the extraocular muscles. These six muscles coordinate to regulate eye movements, and the complex interplay of gene expression within them is crucial for maintaining proper ocular motility. Among ocular motility disorders, inferior oblique muscle overaction (IOOA) is particularly notable for its impact on eye movement dynamics. Both primary and secondary IOOA pose unique challenges in understanding their molecular mechanisms.

Primary IOOA is typically congenital and manifests with hyper-elevation and adduction of the eye. It can occur independently or alongside other types of strabismus, but it does not usually present with signs of superior oblique muscle paralysis. In contrast, secondary IOOA often results from ocular movement abnormalities, such as superior oblique muscle paralysis, and is recognized by similar clinical symptoms, treatment approaches, and surgical strategies (*Wallace et al., 2018*; *Gunton, Wasserman & DeBenedictis, 2015*). Previous research has shown increased collagen I, IV, VI, and elastin in primary IOOA, which correlates with muscle stiffness (*Kushner, 2006*). Studies by *Agarwal et al. (2016)* and *Altick et al. (2012)* further highlighted a shift from contractile proteins to extracellular matrix (ECM) regulators in strabismus, underscoring a change from muscle contractility to ECM-driven pathology. *Kushner (2006)* and *Rudell et al. (2020)* differentiated primary IOOA, characterized by chronic muscle shortening and reduced elasticity, from secondary IOOA, marked by hypertrophy due to sustained neural stimulation. However, a comprehensive molecular comparison between these forms is lacking. *Wu et al. (2022)*'s RNA-seq analysis of congenital esotropia and superior oblique palsy (SOP)-associated IOOA identified key molecular differences, such as altered MyHC isoform expression. However, this study's small sample size and narrow focus limit its applicability.

Despite these similarities, the underlying molecular mechanisms may differ between primary and secondary IOOA, suggesting a need for detailed molecular exploration (*Kushner, 2006*; *Parks, 1972*; *Chang & Yang, 1988*; *Mostafa & Kassem, 2018*). Previous studies have highlighted the complexity of IOOA, yet research into the gene expression profiles associated with these conditions remains limited (*Agarwal et al., 2016*; *Altick et al., 2012*; *Wu et al., 2022*; *Meyer, Ludatscher & Zonis, 1984*).

Our study aims to analyze the gene expression profiles of extraocular muscles in both primary and secondary IOOA using advanced molecular techniques. By identifying specific genetic markers and pathways, we aim to enhance understanding of the molecular changes associated with these conditions, which may reflect adaptive responses to altered functional demands. Gaining insights into the genetic underpinnings of IOOA could lead to improved diagnostic precision and therapeutic strategies, ultimately benefiting patients through more targeted interventions and better clinical outcomes.

## MATERIALS & METHODS

### Study design

This retrospective study aimed to investigate the differences in gene expression profiles between primary and secondary IOOA. The study adhered to the principles outlined in the Declaration of Helsinki and was approved by The Medical Ethics Committee of Tianjin Eye Hospital (Approval Number: KY-2023041). We collected extraocular muscle (EOM) tissues from ten patients with primary IOOA and ten patients with secondary IOOA. All participants had undergone inferior oblique muscle myectomy surgery, which involved removal of the temporal muscle portion of the inferior oblique. Written informed consent for EOM biopsies was obtained from all patients or their parents. The collected EOM tissues were immediately frozen in liquid nitrogen for subsequent transcriptome sequencing.

### Data collection

Demographic and clinical data were extracted from the participants' medical records between July 2023 and March 2024, focusing on age, gender, and the degree of IOOA. The diagnosis and grading of IOOA were independently verified by two pediatric ophthalmologists, Dr. Hao R and Dr. Zhang W. Primary IOOA (P group) was diagnosed in the absence of superior oblique muscle palsy and without dysfunction in other extraocular muscles. Secondary IOOA (S group) was diagnosed when inferior oblique overaction occurred secondary to superior oblique muscle palsy, resulting in hypo-infraduction during adduction. IOOA severity was assessed based on vertical deviation during adduction with elevation, categorized as follows: +1 (approximately 10°), +2 (20°), +3 (30°), and +4 (40°), with 0 indicating normal (*Lim et al., 2014*).

### The transcriptome sequencing

RNA sequencing was performed by Biomarker Technologies (Beijing, China). The process included sample preparation, library construction, quality control, and sequencing. RNA quantity and quality were assessed using the Nanodrop 8000 Spectrophotometer (ThermoFisher Scientific, Waltham, USA) and the Agilent 2100 Bioanalyzer/LabChip GXr (Agilent Technologies, Santa Clara, USA). cDNA libraries were prepared following the manufacturer's protocol, and sequencing was carried out using the Illumina NovaSeq 6000 platform.

### Detection of DEGs

Raw sequencing data in fastq format were subjected to quality control using FastQC. Clean reads were then aligned to the human reference genome (hg38) using HISAT2. Gene expression levels were quantified as Fragments Per Kilobase of transcript per Million fragments mapped (FPKM) using StringTie. DEGs were identified using DESeq2 with the following thresholds: |log2 fold change (log2FC)| > 0.58 (corresponding to a fold change of $\geq 1.5$ or $\leq 0.67$) and a false discovery rate (FDR)-adjusted $p$-value $< 0.05$. log2FC is used to measure the expression level differences of DEGs between different groups.

## Enrichment analysis

Gene Ontology Consortium annotation was enriched using ClusterProfiler. The *P*-value was adjusted by the Benjamini and Hochberg correction, and GO terms with a false discovery rate (FDR) < 0.05 were considered to be significantly enriched. GO annotation system contains three main branches: Biological Process (BP), Molecular Function (MF) and Cellular Component (CC). Enrichment analyses for BP, CC and MF were performed using ClusterProfiler and ClueGo. The terms obtained from the enrichment results were visualised chord plots and composition ratio diagram. GSEA was processed on all genes based on expression level. Enriched gene sets were identified as FDR < 0.05. Additionally, the PPI network was built based on the DEGs generated in the differential expression analysis. The PPI networks were visualized by STRING (https://string-db.org/).

## RESULTS

### Demographic characteristics of the sample

A total of 20 temporal inferior oblique specimens were meticulously collected, comprising ten specimens from patients with primary inferior oblique overaction (IOOA) and ten from patients with secondary IOOA. The mean age of individuals with primary IOOA was 4.4 ± 2.5 years, while those with secondary IOOA had a mean age of 4.2 ± 2.3 years. No statistically significant difference in age was observed between the two groups. Gender distribution was similar, with 50% of patients in the primary IOOA group being female and 60% in the secondary IOOA group, though this difference was not statistically significant. Additionally, there was no significant difference in the degree of IOOA between the two groups (Table 1).

All data were analyzed using SPSS 25.0 (IBM Corp., Armonk, NY, USA). Continuous data with a non-normal distribution were presented as median (Q1, Q3) and analyzed using the Mann–Whitney U test. Two-sided *p* values < 0.05 were considered statistically significant.

### Sample identification and differential gene expression in primary group *vs.* secondary group

A total of 20 samples were processed for transcriptome sequencing, generating 145.66 Gb clean data. At least 6.31 Gb clean data were generated for each sample with minimum 92.50% of clean data achieved quality score of Q30. Clean reads of each sample were mapped to specified reference genome. Mapping ratio ranged from 94.83% to 95.96%. The statistical power of this experimental design, calculated in RNASeqPower was 0.86.

A total of 26,675 genes were identified in the EOM samples. Applying a cutoff of *p*-value < 0.05 and fold change of $\geq$ 1.5 or $\leq$ 0.67, we identified 258 differentially expressed genes (DEGs). Among these, 148 genes were significantly downregulated, while 110 were upregulated in the primary IOOA group compared to the secondary IOOA group (Fig. 1; Table S1). Volcano plots and heatmaps illustrated distinct gene expression patterns between the primary and secondary groups.

**Table 1  Demographic characteristics of the sample.** All data were analyzed using SPSS 25.0. Continuous data with an non-normal distribution were presented as median (Q1, Q3) and analyzed using the Mann–Whitney U test. Two-sided *p* values < 0.05 were considered statistically significant.

|  | Primary IOOA | Secondary IOOA | *p* value |
|---|---|---|---|
| Sex, n (%) |  |  |  |
| Male | 5 (50.0) | 4 (40.0) | 0.661 |
| Female | 5 (50.0) | 6 (60.0) |  |
| Age, years |  |  |  |
| Medium (Q1, Q3) | 3.50 (2.75,6.50) | 4.00 (2.00,6.25) | >0.999 |
| Degree of inferior oblique overaction, n (%) |  |  |  |
| +1 | 0 | 0 |  |
| +2 | 4 (40.0) | 4 (40.0) | 0.796 |
| +3 | 6 (60.0) | 5 (50.0) |  |
| +4 | 0 | 1 (10.0) |  |

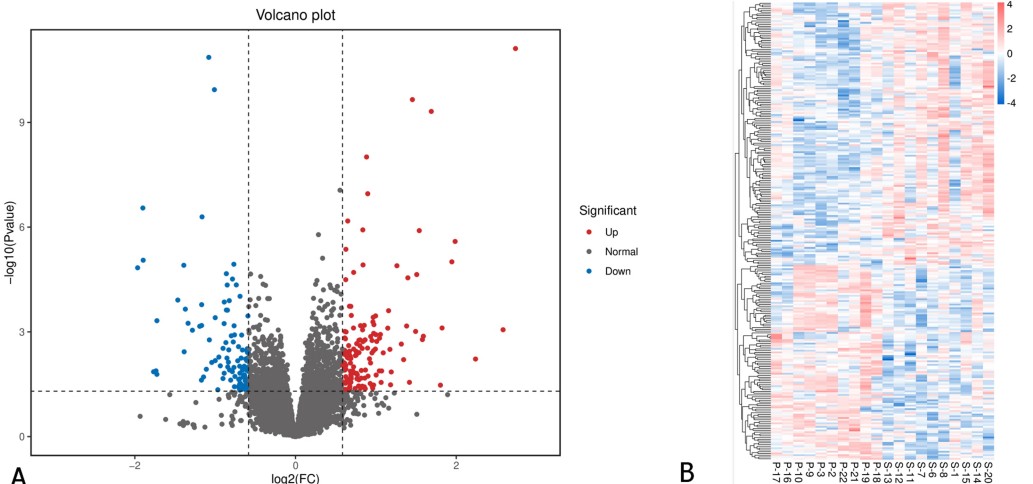

**Figure 1  Differentially expressed genes in EOMs of primary inferior oblique overaction comparing with secondary inferior oblique overaction.** (A) Differentially expressed genes were identified with the thresholds of *p* value < 0.05 and log2FC > 0.58. Red dots represent up-regulated genes, and green ones indicate down-regulated genes. (B) Hierarchical clustering analysis of differentially expressed genes. Higher abundance is colored in red, the lower ones in blue. P, primary inferior oblique overaction; S, secondary inferior oblique overaction.

## Biological pathways enrichment analysis

Enrichment analysis revealed that upregulated DEGs in the primary IOOA group were predominantly associated with immune responses, including positive regulation of chemokine production, regulation of T cell migration, and myelination (*e.g.*, myelination in the peripheral nervous system). Additionally, pathways related to oxygen carrier activity were enriched. Conversely, downregulated DEGs were associated with ion channels (*e.g.*, atrial cardiac muscle cell action potential), negative regulation of signaling receptor activity,

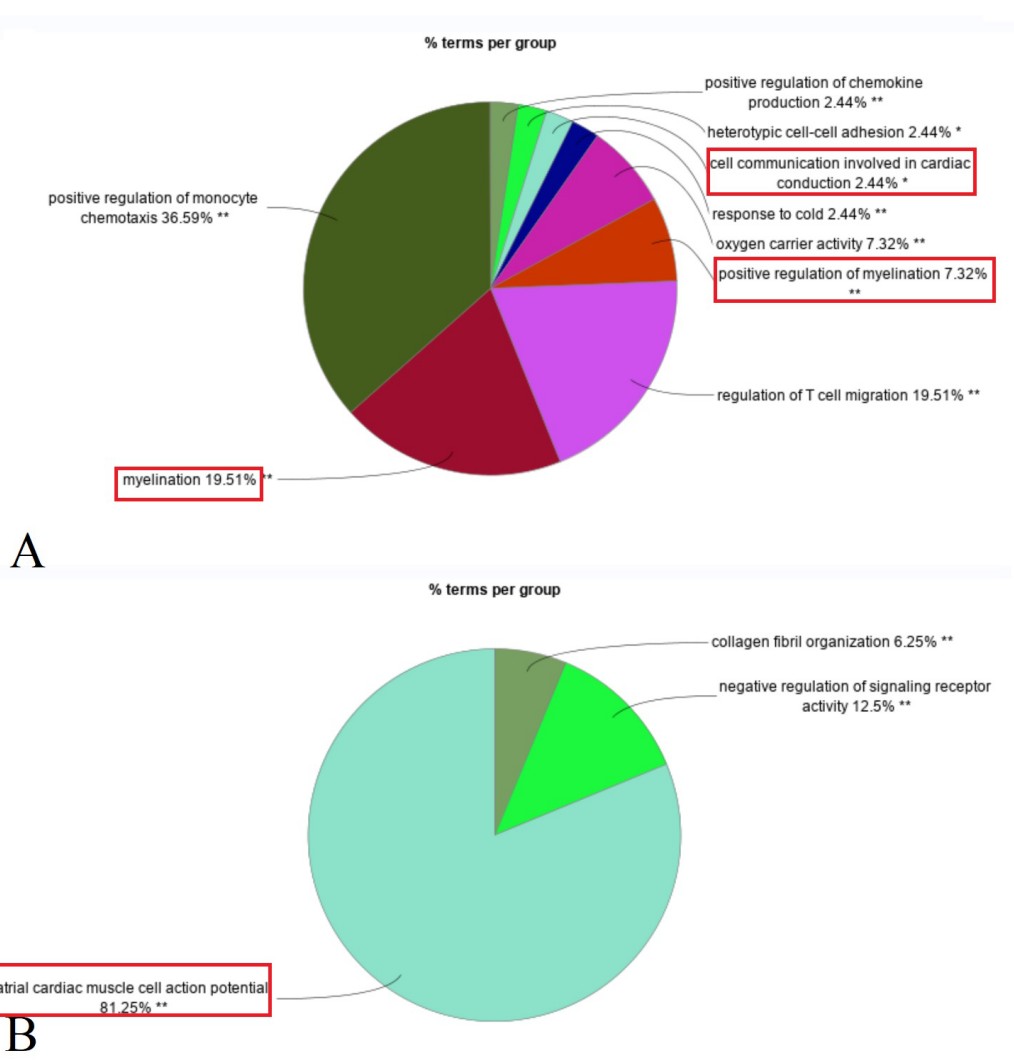

**Figure 2 Composition ratio diagram of enriched GO biological pathways (Red rectangle: GO terms related to the structure and function of nerves and muscles).** (A) Up-regulated DEGs in primary group enriched in myelination, positive regulation of myelination, cell communication involved in cardiac conduction (ion channels). (B) Down-regulated DEGs in primary group enriched in atrial cardiac muscle cell action potential (ion channels).

and collagen fibril organization (Fig. 2). A detailed list of the specific DEGs associated with each enriched GO biological pathway is compiled in the Table S2.

## Cellular components and molecular function enrichment analysis

Analysis of cellular components showed that DEGs in the primary group were enriched in M-bands (*e.g.*, muscle myosin complex, myosin II complex, and A band), ion channels (*e.g.*, voltage-gated potassium and sodium channel complexes), hemoglobin complex, chylomicron, recycling complex, and perinuclear chromocenter (Fig. 3A, Table 2).

Molecular function enrichment analysis indicated that DEGs were significantly associated with muscle alpha-actinin binding, cytoskeletal protein binding, structural

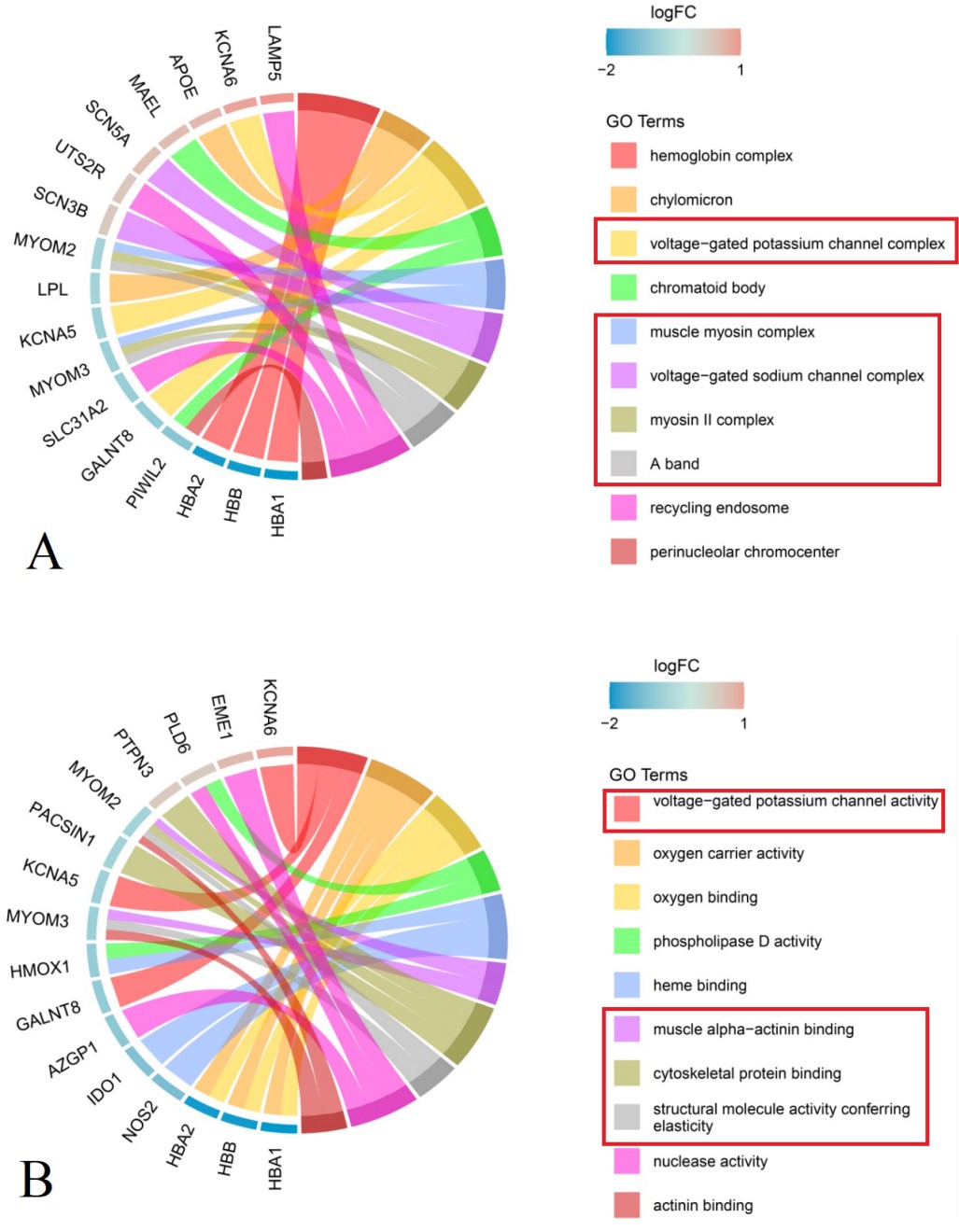

**Figure 3** **Chord plot of the top 10 enriched Gene Ontology (GO) terms (P group *vs* S group, Red rectangle: GO terms related to the structure and function of nerves and muscles).** (A) Cellular Component MYOM2 and MYOM3 were involved in M-band (muscle myosin complex, myosin II complex, A band); SCN3B, SCN5A, KCNA5 and KCNA6 were involved in ion channels (voltage-gated potassium channel complex, voltage-gated sodium channel complex). (B) Molecular Function MYOM2 and MYOM3 were involved in M-band (muscle alpha-actinin binding, 

**Figure 3 (…continued)**
cytoskeletal protein binding, structural molecule activity conferring elasticity); KCNA5 and KCNA6 were involved in ion channels (voltage-gated potassium channel complex). Note: The left side of the outer circle represented differential genes, the red block represented upregulated genes, and blue represented down-regulated genes. Genes were ordered according to the observed log-fold change (logFC), which was displayed in descending intensity of red or blue squares displayed next to the selected genes. On the right side of the outer circle was GO terms, with different colors representing different terms. The genes were linked to their assigned terms *via* colored ribbons.

**Table 2  Full names of DEGs enriched in cellular components and molecular function.**

| Gene abbreviation | Gene ID | Full names of genes |
|---|---|---|
| APOE | 348 | Apolipoprotein E |
| AZGP1 | 563 | Alpha-2-glycoprotein 1, zinc-binding |
| EME1 | 146956 | Essential meiotic structure-specific endonuclease 1 |
| GALNT8 | 26290 | Polypeptide N-acetylgalactosaminyltransferase 8 |
| HBA2 | 3040 | Hemoglobin subunit alpha 2 |
| HBB | 3043 | Hemoglobin subunit beta |
| HMOX1 | 3162 | Heme oxygenase 1 |
| IDO1 | 3620 | Indoleamine 2,3-dioxygenase 1 |
| KCNA5 | 3741 | Potassium voltage-gated channel subfamily A member 5 |
| KCNA6 | 3742 | Potassium voltage-gated channel subfamily A member 6 |
| LAMP5 | 24141 | Lysosomal associated membrane protein family member 5 |
| LPL | 4023 | Lipoprotein lipase |
| MAEL | 84944 | Maelstrom spermatogenic transposon silencer |
| MYOM2 | 9172 | Myomesin 2 |
| MYOM3 | 127294 | Myomesin 3 |
| NOS2 | 4843 | Nitric oxide synthase 2 |
| PACSIN1 | 29993 | Protein kinase C and casein kinase substrate in neurons 1 |
| PIWIL2 | 55124 | Piwi like RNA-mediated gene silencing 2 |
| PLD6 | 201164 | Phospholipase D family member 6 |
| PTPN3 | 5774 | Protein tyrosine phosphatase non-receptor type 3 |
| SCN3B | 55800 | Sodium voltage-gated channel beta subunit 3 |
| SCN5A | 6331 | Sodium voltage-gated channel alpha subunit 5 |
| SLC31A2 | 1318 | Solute carrier family 31 member 2 |
| UTS2R | 2837 | Urotensin 2 receptor |

molecule activity conferring elasticity, and actinin binding within M-bands. In ion channels, the voltage-gated potassium channel complex was notably enriched. Furthermore, genes related to oxygen carrier activity demonstrated enrichment in oxygen carrier activity, oxygen binding, heme binding, as well as phospholipase D activity and nuclease activity (Fig. 3B, Table 2).

## GSEA and PPI analysis validation
GSEA and PPI network analysis further validated the above results of the enrichment analysis of biological pathways, cellular components, and molecular functions. Additionally,
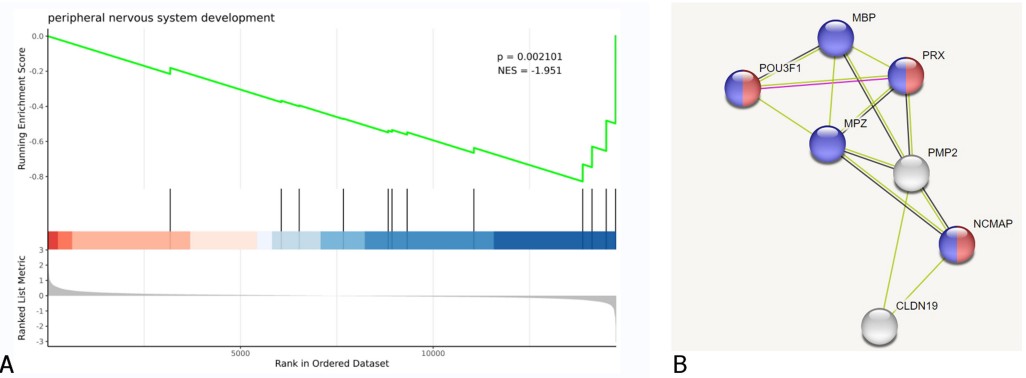

**Figure 4** **DEGs were clustered in peripheral nervous system development.** (A) Biological process of GSEA. (B) A cluster of constructed PPI network of DEGs using STRING Red nodes: Myelin sheath, a cellular component term; Blue nodes: Myelination, a biological process term.

it was discovered that some DEGs are associated with peripheral nervous system development and myelin sheath formation (Fig. 4, Table 2).

## GO terms related to nerves, muscles and extracellular matrix

Gene Ontology (GO) analysis identified several biological terms linked to nerves and muscles, such as myelination, ion channels, M-bands, and collagen fibril organization. In particular, key DEGs related to myelination, including MBP, MPZ, and PRX, showed elevated expression in the primary IOOA group compared to the secondary IOOA group. Ion channel-related DEGs, such as KCNA5 and KCNE5, were also upregulated, while genes like CACNA1B, SCN3B, SCN5A, and KCNJ3 exhibited a decrease in expression. Additionally, DEGs associated with M-bands, including MYOM2 and MYOM3, were more highly expressed in the primary group than in the secondary IOOA group. On the other hand, DEGs related to collagen fibril assembly, including COL11A1, COL11A2, COL22A1, COL25A1, and COL9A2, were found to be upregulated in the secondary IOOA group (Table 3).

## DISCUSSION

In this study, we performed transcriptome sequencing to delineate the distinctive gene expression profiles of EOMs in primary *versus* secondary inferior oblique overaction (IOOA). We identified 258 DEGs, with 110 showing significant upregulation and 148 downregulation in the primary IOOA group. Our bioinformatics analysis revealed enrichment of DEGs in neuromuscular processes, notably myelination, ion channels, and M-bands and collagen fibril. These findings suggest a higher prevalence of multiply innervated fibers (MIFs) in the primary IOOA group, which likely contributes to the observed heightened slow muscle tension, consistent with the reported increased nerve fiber density and increased density of neuromuscular junctions in inferior muscle with primary overaction (*Rudell et al., 2020*). In addition, the elevated presence of collagen

**Table 3** The filtered DEGs related to nerves and muscles (P group *vs.* S group).

| Gene class | Gene symbol | Full name | log2FC | *P* value | Regulated |
|---|---|---|---|---|---|
| Myelination | MBP | Myelin basic protein | −0.592409143 | 0.000350453 | Up |
| | MPZ | Myelin protein zero | −1.192710164 | 0.000692698 | Up |
| | PRX | Periaxin | −0.824815977 | 0.000127739 | Up |
| Ion channels | CACNA1B | Voltage-dependent N-type calcium channel subunit alpha-1B | 0.654443519 | 0.025644469 | Down |
| | SCN5A | Sodium voltage-gated channel alpha subunit 5 | 0.788437111 | 0.001747162 | Down |
| | SCN3B | Sodium voltage-gated channel beta subunit 3 | 0.625062455 | 3.20E−05 | Down |
| | KCNJ3 | G protein-activated inward rectifier potassium channel 1 | 1.261998049 | 1.28E−05 | Down |
| | KCNA5 | Potassium voltage-gated channel subfamily A member 5 | −0.661188331 | 0.012416218 | Up |
| | KCNE5 | Potassium voltage-gated channel subfamily E regulatory beta subunit 5 | −0.679199276 | 0.032395393 | Up |
| M-bands | MYOM2 | Myomesin 2 | −0.587331343 | 0.013452704 | Up |
| | MYOM3 | Myomesin 3 | −0.677021812 | 0.005638619 | Up |
| Collagen fibril | COL11A1 | Collagen type XI alpha 1 chain | 1.41806248 | 0.027906927 | Down |
| | COL11A2 | Collagen type XI alpha 2 chain | 1.056676321 | 0.001184297 | Down |
| | COL22A1 | Collagen type XXII alpha 1 chain | 0.954426441 | 0.036265677 | Down |
| | COL25A1 | Collagen type XXV alpha 1 chain | 0.898696538 | 1.10E−07 | Down |
| | COL9A2 | Collagen type IX alpha 2 chain | 0.613171134 | 0.001493233 | Down |

**Notes.**
Fold change of ≥ 1.5 or ≤ 0.67 and false discovery rate (FDR) Benjamini–Hochberg-adjusted *P*-value < 0.05 were used as screening criteria for detection of DEGs using DESeq2.

fibrils, which may contribute to muscle stiffness in the secondary IOOA group, might be related to compensatory mechanisms in this group (*Agarwal et al., 2016*; *Altick et al., 2012*).

## Gene expression patterns in myelination

The upregulation of myelination-related genes in the primary IOOA group is particularly noteworthy. Genes such as MBP (myelin basic protein), MPZ (myelin protein zero), and PRX (peripheral myelin protein 22) were significantly upregulated. MBP is essential for the stability and maintenance of the myelin sheath (*Simons & Trotter, 2007*), MPZ facilitates myelin-axon adhesion (*Lagueny et al., 1999*), and PRX is crucial for Schwann cell integrity and myelin repair (*Gillespie et al., 1997*). The increased expression of these genes implies a higher prevalence of MIFs in the primary IOOA group, which are known for their role in sustaining prolonged muscle contractions (*Rudell et al., 2020*).

In extraocular muscles, MIFs differ from singly innervated fibers (SIFs) by their unique innervation patterns and functional properties. MIFs, characterized by en grappe terminals, generate slow, sustained tension rather than rapid twitches, which is critical for precise eye movements and gaze stabilization (*Hernández et al., 2019*). This distinction is supported by our finding of increased myelination-related gene expression in the primary IOOA group.

Furthermore, it is noteworthy that the vascular-ligamentous-nerve strand, unique to the inferior oblique muscle, is a critical anatomical feature that could offer valuable insights into the differential gene expression observed in this study. The interactions between the vascular, ligamentous, and neural components could influence the muscle's response to abnormal functional demands, possibly leading to compensatory adaptations such as

myelination or other molecular changes. Stager et al. described this unique structure, which comprises a combination of blood vessels, ligaments, and nerve fibers, and serves as a conduit for neuromuscular communication and vascular support to the inferior oblique muscle (*Stager Jr, McLoon & Felius, 2013*; *Stager Jr, Dao & Felius, 2015*). This structure may play a role in regulating the muscle's response to various functional and mechanical demands, which could, in turn, influence gene expression patterns.

## Dysregulation of ion channels

Our analysis revealed dysregulation of ion channel genes in the primary IOOA group. Specifically, downregulation was observed in genes associated with calcium channels (CACNA1B), sodium channels (SCN5A, SCN3B), and potassium channels (KCNJ3), while upregulation was noted for genes encoding potassium channels (KCNA5, KCNE5). Ion channels are vital for the generation and propagation of action potentials in muscle fibers (*Phillips & Trivedi, 2018*). The downregulation of CACNA1B, which encodes N-type calcium channels, may lead to reduced neurotransmitter release and diminished endplate potential amplitude (*Nudler et al., 2005*). Similarly, the downregulation of SCN5A and SCN3B, which are crucial for sodium channel function, could result in decreased sodium influx and impaired action potential initiation (*Fahmi et al., 2001*).

Conversely, the upregulation of KCNA5 and KCNE5 suggests faster repolarization, which aligns with the characteristics of MIFs that require rapid adaptation to repetitive stimulation. These ion channel alterations reflect the unique electrical properties of MIFs, which are adapted to maintain slow, sustained tension.

*KCNJ3* is associated with allowing potassium influx into cells, crucial for regulating heartbeat (*Fleischmann et al., 2004*), while *KCNA5* contributes to restoring the resting membrane potential after depolarization, impacting insulin secretion regulation (*López-Vera et al., 2020*). *KCNE5* acts as an ancillary subunit to voltage-gated potassium channels (*Abbott, 2016*). The down-regulation of *KCNJ3* and up-regulation of *KCNA5* and *KCNE5* in the primary IOOA group may suggest faster repolarization.

## M-band gene alterations

The upregulation of M-band-related genes, MYOM2 and MYOM3, in the primary IOOA group is also significant. The M-band, located at the center of the sarcomere, plays a critical role in managing force distribution during muscle contraction (*Lange et al., 2020*). MYOM2 and MYOM3 are integral to the structural integrity and functional performance of the M-band. MYOM2, predominantly expressed in fast skeletal muscles, and MYOM3, found in slow muscle fibers and extraocular muscles, contribute to muscle stiffness and resistance to fatigue (*Wiesen et al., 2007*; *Schoenauer et al., 2008*). In the context of muscle stiffness, the findings of previous studies are particularly relevant (*Kushner, 2006*; *Agarwal et al., 2016*; *Altick et al., 2012*; *Stager Jr, McLoon & Felius, 2013*; *Stager Jr, Dao & Felius, 2015*). These studies have highlighted how changes in muscle stiffness, such as those observed in overacting extraocular muscles, may reflect underlying structural and functional alterations, including increased innervation and fiber adaptation. This framework aligns with our observation of increased MYOM2 and MYOM3 expression, potentially contributing to the

heightened muscle tension in primary IOOA. The increased expression of these genes in the primary IOOA group may enhance muscle rigidity and fatigue resistance, supporting the unique contractile properties of extraocular muscles in this condition.

## Expression changes of genes related to skeletal muscle contraction and elasticity

The contraction and elasticity of skeletal muscle are related to various proteins within its structure. Among them, actin and myosin are the core proteins responsible for skeletal muscle contraction, and their interaction generates force through the cross-bridge cycle. Titin is one of the most important elastic proteins in muscle, providing elastic recoil during passive stretching and preventing overstretching. Collagen fibers in the extracellular matrix (ECM) also contribute to the passive elasticity of muscle, particularly during stretching (*Roberts, 2016*).

Our study found no significant differences in the transcriptional levels of key proteins related to muscle contraction, such as actin and myosin, between the primary and secondary groups. However, *Wu et al. (2022)*'s RNA-seq analysis of congenital esotropia and SOP-associated inferior oblique overaction (IOOA) identified upregulation of three fast-twitch myosin heavy chain (MyHC) isoforms (MYH1, MYH4, and MYH13) and downregulation of one slow-twitch MyHC isoform (MYH3) in the SOP group. This may suggest a strengthening of the inferior oblique muscle in secondary IOOA. However, *Wu et al. (2022)*'s study included only five cases of SOP-associated IOOA and three cases of congenital esotropia, which limits the sample size, and these findings require further validation. The differences between our study and *Wu et al. (2022)*'s may be attributed to variations in sample characteristics. Therefore, additional research with a larger sample size is necessary to confirm these results.

We investigated the expression of genes related to muscle stiffness and found that collagen genes, including COL11A1, COL11A2, COL22A1, COL25A1, and COL9A2, play crucial roles in the structure and function of collagen fibers, vital for connective tissues, myotendinous junction stabilization, and cartilage integrity (*Annunen et al., 1999*; *Melkoniemi et al., 2000*; *Koch et al., 2004*; *Holden et al., 1999*). However, other stiffness-related genes, including titin, actin, and myosin, did not show significant differences between primary and secondary IOOA. The observed reduction in collagen gene expression in the primary IOAA group was unexpected and surprising, given that prior studies have consistently demonstrated elevated collagen content in the inferior oblique muscle of primary IOAA patients (*Stager Jr, McLoon & Felius, 2013*). This discrepancy might imply that the secondary IOOA group exhibits elevated muscle stiffness, potentially attributable to compensatory adaptations. However, due to the absence of normal controls in our study, we cannot determine the expression changes of stiffness-related genes in the primary group.

## Clinical implications and future directions

The distinct gene expression profiles observed in primary *versus* secondary IOOA provide valuable insights into the underlying mechanisms of these conditions. The increased

prevalence of MIFs and the dysregulation of ion channels and M-band proteins in primary IOOA may contribute to the unique clinical features of this disorder. These findings offer potential targets for therapeutic interventions and highlight the need for further research to explore the functional implications of these DEGs.

## Study limitations and future research

Despite the valuable insights gained, this study has several limitations. First, the absence of a baseline gene expression for normal inferior oblique muscle in this study limits the interpretation of gene expression changes. Furthermore, the sample size is adequate for preliminary findings, it may not fully reflect the variability within the inferior oblique overaction (IOOA) population. Moreover, the cross-sectional design of the study restricts the ability to establish causative relationships between observed gene expression changes and clinical outcomes. It is important to note that gene expression changes in overacting inferior oblique muscles may not be causative but rather compensatory, potentially reflecting the muscle's adaptation to altered functional demands. This aligns with previous studies (*Wu et al., 2022*; *Parks, 1972*), which emphasize the plasticity of extraocular muscles in response to biomechanical and neural stimuli. To further validate these findings and explore the functional implications of DEGs, future research should include larger cohorts and adopt longitudinal designs. Additionally, incorporating functional assays and molecular imaging techniques will provide a more detailed understanding of how genetic alterations influence muscle function and ocular motility.

### Funding

The present study was supported by General Project of Tianjin Health Science and Technology Fund (grant no. TJWJ2021MS041), Tianjin Key Medical Discipline (Specialty) Construction Project (grant no. TJYXZDXK-016A), Tianjin Eye Hospital Science and Technology Fund Youth Cultivation Project (grant no. YKPY2201), the open project of Institute of Optometry and Vision Science in Nankai University (NKSGY202309). The funders had no role in study design, data collection and analysis, decision to publish, or preparation of the manuscript.

### Grant Disclosures

The following grant information was disclosed by the authors:
General Project of Tianjin Health Science and Technology Fund:  TJWJ2021MS041.
Tianjin Key Medical Discipline (Specialty) Construction Project: TJYXZDXK-016A.
Tianjin Eye Hospital Science and Technology Fund Youth Cultivation Project:  YKPY2201.
The open project of Institute of Optometry and Vision Science in Nankai University: NKSGY202309.

### Competing Interests

The authors declare there are no competing interests.

## Author Contributions

- Rui Hao conceived and designed the experiments, performed the experiments, analyzed the data, prepared figures and/or tables, authored or reviewed drafts of the article, and approved the final draft.
- Yuchuan Wang conceived and designed the experiments, performed the experiments, analyzed the data, prepared figures and/or tables, authored or reviewed drafts of the article, and approved the final draft.
- Wei Zhang conceived and designed the experiments, performed the experiments, authored or reviewed drafts of the article, and approved the final draft.

## Human Ethics

The following information was supplied relating to ethical approvals (i.e., approving body and any reference numbers):

This retrospective study received approval from ethics committee of Tianjin Eye Hospital and complies with the principles of the Helsinki Declaration and was approved by The Medical Ethics Committee of Tianjin Eye Hospital (Approval Number: KY-2023041).

## Data Availability

The raw data is available at NCBI GEO: GSE281104.

## Supplemental Information

Supplemental information for this article can be found online at http://dx.doi.org/10.7717/peerj.19474#supplemental-information.

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
