# Peer review of "Gene expression profiling of extraocular muscles in primary inferior oblique overaction"

_PeerJ, doi:10.7717/peerj.19474_

## Round 0.1 · original submission · Major Revisions

Both reviewers have identified major issues with the paper, and it is unclear whether all of these issues can be addressed. If the authors choose to revise the paper, please address every comment.

Reviewer 1 ·

Basic reporting

Professional English is used, well written
Background Literature has some significant gaps that need to be filled
The article is professionally structured (some minor edits are needed in legends)
See ADDITIONAL COMMENTS for details

Experimental design

Could use major improvement to be truly relevant
Technically adequate, but some important information about muscle samples is missing, so that it can be replicated
No ethics issues or concerns
See ADDITIONAL COMMENTS for details

Validity of the findings

The supplemental data could be made more “user-friendly”
Limitations of the experimental design should be stated more clearly (including that gene expression changes may not be causative, and that pathogenesis is not explored – contrary to what is said in the Introduction)
See ADDITIONAL COMMENTS for details

Additional comments

This paper identifies differentially expressed genes in the inferior oblique muscle by comparing gene expression when the muscle is primarily overacting with muscle that is overacting due to secondary causes. The authors find differences in gene expression mostly in genes involved in myelination and ion channels.

While the technical details are convincing and the study is novel, I do have serious concerns about this study.
1. Foremost, the authors do not compare gene expression with a baseline of a normal inferior oblique muscle. This severely limits the conclusions that can be made, because all those genes which are similarly up-regulated or down-regulated when compared with a normal muscle will not be visible when only two types of overacting inferior oblique muscle are compared – which is the authors’ experimental strategy in this study. This likely is the reason why they do not find any or very few genes related to muscle stiffness (collagen, extracellular matrix, elastin, etc), as has been consistently reported in strabismic inferior oblique muscles and strabismic horizontal muscles based on previous work.
2. I realize that it is not easy to obtain suitable normal extraocular muscles for comparison. To obtain such normal muscles, previous studies have used cases of enucleation - resection due to tumors or other conditions (Stager et al., 2013; Rudell et al., 2020), while another group has used tissue from organ donors with no history of eye muscle dysfunction (Altick et al., 2012; Agarwal et al., 2016).
3. It needs to be explained that gene expression changes may not be causative to the overaction condition, because they may be compensatory, due to the muscle’s effort to adjust to abnormal functional demands (See, e.g., Altick et al., 2012; Agarwal et al., 2016). This changes dramatically how the gene expression data are interpreted.
4. Did the resected samples contain only muscle or also tendon? Gene expression changes in strabismic muscles have been shown to differ substantially between muscle and tendon (Agarwal et al., 2016). Also, was the entire muscle included in the samples or only parts thereof? E.g. temporal/nasal components (Stager et al., 2013) which has been described as a confounding variable (Rudell et al., 2020). This should be specified and should be similar in the muscle samples from primary and secondary overaction conditions. Significant gradients in gene and protein expression have also been described along the longitudinal axis of human extraocular muscles (Agarwal et al., 2016).
5. The study misses some of the most relevant papers on gene and protein expression changes in human extraocular muscles, and even some studies specifically on overacting inferior oblique muscles. There should be a much more comprehensive analysis and discussion on this than what is currently offered.
6. Discuss the vascular-ligamentous-nerve strand which is unique to inferior oblique muscle (Stager et al., 2013; Stager et al., 2015), as this may explain some of the differential gene expression described by the authors (e.g., myelination).
7. Kushner (2006) postulated a loss of elasticity in primary IOOA, but a truly strengthened inferior oblique muscle in secondary IOOA. Does the gene expression profile support this notion?

Minor points:
Lines 55-56: It is true that work on gene expression profiles of inferior oblique muscle has been limited, but none of the 4 references offered (#6-9) deals with gene expression – they all report on structural or cellular changes. Gene expression studies such as Ref. #14, or studies on altered gene expression in horizontal extraocular muscles (Altick et al., 2012; Agarwal et al., 2016) would seem more appropriate in this context.
Line 58: “understanding of the pathogenesis” – the reader should be advised that gene expression in overacting muscles may not inform about the pathogenesis but may rather reflect compensatory changes of the muscle to altered functional demands.
Lines 67, 70: There should be more detailed information about the “muscle tissues” – was the entire EOM available? Was only muscle examined, or also tendon? Were both temporal and nasal components included? Were the samples comparable in muscle and tendon content between the primary overaction condition vs. secondary overaction condition?
Line 76: how was the dysfunction in other extraocular muscles determined? Also, it is surprising that for primary IOOA all cases were excluded in which other extraocular muscles were abnormal, since IOOA often is associated with horizontal strabismus (Stager et al., 2013). Were cases excluded even when the infantile esotropia had been corrected years earlier? Please clarify.
Line 104: Again – was the entire muscle examined in each of the samples?
Line 112: “data with an …” – “data with a …”
Line 117: 20 … – 20 samples …
Line 129: Production of which chemokines was positively regulated? Name the most important examples?
Line 133: “collagen fibril organization” – more specific information about the relevant genes is desirable here, because collagen genes and proteins were reported to be upregulated/altered in strabismic EOMs (Altick et al., 2012; Agarwal et al., 2016, and references cited in those reports: Martinez et al., 1980; Kim et al., 2008)
Line 141: Since it has been described that IOOA may be due primarily to changes in muscle stiffness and changes in collagen, extracellular matrix, and elasticity (Stager et al., 2013; Stager et al., 2015), additional information on the gene/molecules conferring elasticity should be given here.
Line 148: “revealing additional insights” – what are those insights?
Line 153-154: I suggest to add at the end of the sentence “… relative to the secondary IOOA group.” (to make it clear that this is not a comparison with the normal IO).
Lines 214-216: In this context, it should be stated that these findings are consistent with the reported increased nerve fiber density and increased density of neuromuscular junctions in inferior muscle with primary overaction (Rudell et al., 2020).
Lines 222-223: “implies a higher prevalence of MIFs” – again, this was shown by Rudell et al., 2020 and should be cited. The reference #14 does not seem suitable in this context.
Line 251: In the context of muscle stiffness, the work of Kushner, 2006; Stager et al., 2013, 2015, and Altick et al., 2012; Agarwal et al., 2016 should be discussed.
Line 264: “longitudinal design” should be explained – do the authors envision examination of gene expression at different ages, or at different time periods after development of IOOA? In this context, it is important to note that comparison of EOMs from different ages (Agarwal et al., 2016) suggests that age may not be a major relevant factor.
Lines 265-267: The major limitation of the study in its current form is the lack of a baseline gene expression in a normal inferior oblique muscle. If the authors do not add such a crucial third group, then this should be explained and justified.
Figure 2 legend: explain what * and ** mean. There should also be some explanation – in the text and/or legend – how genes related to the atrial cardiac muscle action potential is relevant for extraocular muscle function.
Figure 3 legend: It would be useful to provide a short table that explains (give full names) of all the genes illustrated in panels A and B.
Figure 4 legend: Insert “.” after STRING. Insert a “;” after “system”. Also, explain the gene names in panel B.

Suggested Reading and Consideration:

Kushner BJ. Multiple mechanisms of extraocular muscle "overaction". Arch Ophthalmol. 2006 May;124(5):680-8. doi: 10.1001/archopht.124.5.680.
Altick AL, Feng CY, Schlauch K, Johnson LA, von Bartheld CS. Differences in gene expression between strabismic and normal human extraocular muscles. Invest Ophthalmol Vis Sci. 2012 Aug 3;53(9):5168-77. doi: 10.1167/iovs.12-9785.
Stager D Jr, McLoon LK, Felius J. Postulating a role for connective tissue elements in inferior oblique muscle overaction (an American Ophthalmological Society thesis). Trans Am Ophthalmol Soc. 2013 Sep;111:119-32.
Stager D Jr, Dao LM, Felius J. Uses of the Inferior Oblique Muscle in Strabismus Surgery. Middle East Afr J Ophthalmol. 2015 Jul-Sep;22(3):292-7. doi: 10.4103/0974-9233.159723.
Agarwal AB, Feng CY, Altick AL, Quilici DR, Wen D, Johnson LA, von Bartheld CS. Altered Protein Composition and Gene Expression in Strabismic Human Extraocular Muscles and Tendons. Invest Ophthalmol Vis Sci. 2016 Oct 1;57(13):5576-5585. doi: 10.1167/iovs.16-20294.
Rudell JC, Stager D Jr, Felius J, McLoon LK. Morphological Differences in the Inferior Oblique Muscles from Subjects with Over-elevation in Adduction. Invest Ophthalmol Vis Sci. 2020 Jun 3;61(6):33. doi: 10.1167/iovs.61.6.33.

·

Basic reporting

good paper

Experimental design

its perfect and good information

Validity of the findings

no comment

Additional comments

Reviewer Comments:
The study investigates gene expression differences in primary inferior oblique overaction (IOOA) by performing transcriptome sequencing on extraocular muscles (EOMs) from patients with primary and secondary IOOA. The study evaluates outcomes related to visual acuity, refractive errors, and patient symptoms, in primary inferior oblique overaction (IOOA).
1. The study provides valuable short-term data, but it would benefit from including a longer follow-up period to assess the durability and stability of the outcomes.
2. The study focuses on primary inferior oblique overaction (IOOA) by performing transcriptome sequencing on extraocular muscles (EOMs) from patients with primary and secondary IOOA. but omits key ocular parameters such as high-order aberrations, and the relationship between clinical traits and histopathological as well as immunohistochemical characteristics of inferior oblique muscles in individuals experiencing primary and secondary inferior oblique overaction, you are using bioinformatic analyses to identify differentially expressed genes (DEGs) and enriched pathways.
3. The abstract did not mention an introduction to strabismus and genes, nor did the research methodology mention the method you used and the period you used to collect the data. As for the results, you did not mention the most important results you achieved. They should be added to the abstract.
4. Your introduction requires additional detail. I recommend enhancing the description in lines 48-54 to offer more justification for your research (in particular, you should elaborate on the knowledge gap being addressed).
5. The English in the writing is good and understandable but in lines 52-54 it should be rephrased to ensure that an international audience can understand your text clearly. You should add details and define more.
6. The enrichment analysis in lines 96-101 was not understandable and the analysis method was not able to add details and words that the reader can understand and benefit from your research in the future.
7. The method was good, but you did not mention consent or a picture of how to obtain informed consent for EOM biopsies from all patients. Also, the time period for data collection and analysis was short and the sample size of participants was also small. You should add it to your paper. In the feature, you should specify small to large sizes. Finally, you didn’t mention the participants. Finally, you did not mention how the participants had or got IOOA and secondary IOOA due to heredity or accident or trauma because the ages of the participants were (2.5-6.5) years. mention that in your paper.
8. The statistical analysis section is somewhat brief and could be improved by explaining in more depth the choice of tests, including assumptions made and whether tests for normality were conducted before applying for parametric tests.
9. Tables and figures are useful but could be more explicit. The labels on figures (e.g., Figure 1) could be more detailed, and tables may benefit from additional explanations of statistical significance results to make them easier for readers to interpret. Details should also be added below the table, such as the type of analysis or SPSS method used.
10. For reference, recent sources from 2012 and above must be cited to be consistent with the journal.
11. Please consider citing Wu X,et all, 2022 May 1;11(5):676-86.

In conclusion, the manuscript suffers from some errors that need to be corrected. The topic of profiling extraocular muscle gene expression in primary inferior oblique hyperactivity is not new but is important. There are several publications on this topic. However, external studies have revealed molecular distinctions that may underline the unique neuromuscular characteristics of primary inferior oblique hyperactivity. As the authors describe the results on this new platform, this topic may be of interest to readers. However, some rewording of sentences is needed as per the reviewers’ comments for this paper that address all the issues to make this publication worthwhile.

---

## Round 0.2 · Minor Revisions

Thank you for your previous reviewer responses to the reviewer comments. A reviewer has re-reviewed the paper and has requested some specific additions/changes. Please address each of these comments.

Reviewer 1 ·

Basic reporting

English language needs very few, very minor edits.
Literature, context and background now are adequately provided.
Some figures need a few additional explanations, but they are largely ok.
Raw data are shared with a start date of 2028.

Experimental design

The research is within the aims and scope of the journal.
The research question is well defined, and limitations are discussed.
The investigation is technically sound.
No ethics concerns.
Methods are sufficiently explained.

Validity of the findings

Rationale and benefit to literature are clearly stated.
Conclusions are limited to supporting results.

Additional comments

The revised version of the manuscript is significantly improved. The authors now place their findings into the appropriate context of previous relevant reports about gene expression in strabismic extraocular muscles. The authors did not change their experimental design by including gene expression in a normal muscle, but such tissue is difficult to obtain, and the authors discuss this limitation in the revised manuscript.

There still are several, mostly very minor, issues that need attention:

Line 57: ECM should be defined
Line 61: SOP should be defined
Line 67: The new sentence should begin on Line 66.
Line 70: I recommend to replace “This” with “Our” – then there is no ambiguity about which study is meant (not the most recently cited one in the previous sentence)
Line 83: I suggest to add “removal of the” after “involved” (just to clarify that the obtained sample was the temporal portion of the muscle)
Line 119: delete “be”
Line 153: “is compiled” seems to be missing in this sentence, after “pathway”
Line 169: “further” can be deleted
Line 264: “Stage et al. described …” – please insert the appropriate reference(s) to clarify whether this is found in reference [17], or [18], or [17,18].
Lines 288-299: In this paragraph, “extraocular muscles” is written out twice (line 292 and 295), but the abbreviation (EOMs) is used on line 299. Be more consistent?
Line 300: the subheading should be on line 301, for consistency.
Lines 317-326: This paragraph is spaced wider than all the other text.
Line 355: In the sequence of discussing study limitations, the absence of a baseline gene expression should be the first one, since it is the most important limitation. Please move the sentence from lines 342-343 to line 335 (after “First”), and then begin the sentence on line 342 about sample size after “Furthermore, …”. (The sample size of 10 subjects for each of the two groups is fairly adequate, so this is a relatively minor limitation).
The references should have a consistent format. The added references [4,5,7,28,29,30,31,32] do not have the volume in bold font, and the year should be after the page numbers in parentheses, not right after the Journal name.
Figure 2 Legend: Go terms – GO terms (for consistency)
Figure 3 Legend: Go terms – GO terms. Also: The caption has two squares (one after “terms”, one after “muscles”). Was the intended symbol (period?) not displayed properly? In addition, the terms “P group” and “S group” should be explained
Figure 4 Legend: “rodes” should be “nodes” (?)
Table 1: Since the explanation of the entries (in the first column) extend over two lines, the actual data in the second, third and fourth column should start one line below where they are now. Also, please explain why there is a p-value for the first line of the degree of overaction, even though there are no samples in that row. For which data does the p-value apply?
Table 2: The caption is duplicated. Also, the Caption announces Table 2, but that Table is (mistakenly?) called Table 3 on the next page.
Table 3: Explain the P group and the S group. Also, the Caption announces Table 3, but that Table is (mistakenly?) called Table 2 on the next page. In addition, some of the entries in that table are in red font, which is not explained. Or was this section mistakenly not converted to black font from the tracked version to the clean version?

---

## Round 0.3 · Minor Revisions

Thank you once again for your responses to the reviewer comments.

Could you please directly address this final issue raised by the reviewer:
"Table 1: Since the explanation of the entries (in the first column) extend over two lines, the actual data in the second, third and fourth column should start one line below where they are now. Also, please explain why there is a p-value for the first line of the degree of overaction, even though there are no samples in that row. For which data does the p-value apply?"

I believe what the reviewer is asking about is why there is a p value adjacent to two "O" values. Does that p value correspond to the overall group of comparisons (+1, +2, +3, +4). Can you please clarify, perhaps in the table caption, what that final p value represents?

---

## Round 0.4 · accepted · Accept

Thank you for addressing all of the reviewer comments and for modifying the table as requested.

Reviewer 1 ·

Basic reporting

no comments

Experimental design

no comments

Validity of the findings

no comments

Additional comments

The concern regarding Table 1 appears to be resolved. There still are a few issues that do need attention. They are easy to fix, but some of them are very important:

Line 29: The results (and Fig. 1A) say that 148 genes were upregulated, and 110 genes were downregulated. The abstract says the opposite – this needs to be corrected.
Line 62: SOP usually stands for superior oblique palsy (not muscle)
Line 84: “and” should be “or” (2-year olds do not sign the written consent)
Line 114: the abbreviations CC and MF are mixed up.
Line 119: was – were
Lines 207-211: “invovled” should be “involved” (4 times)
Lines 224-225: red nodes are defined as myelination in the PNS, and blue nodes as “myelination”. Please explain how the “myelination” differs from the “myelination in the PNS” – does it include both PNS and CNS? Or only CNS?
Lines 241: Again, it says here that 148 genes were downregulated, and 110 were upregulated. It should be the opposite according to the results and Fig. 1A.
Line 247: collagen makes the EOMs more stiff, not more elastic (according to Stager et al [17]). This should be corrected.
Lines 317-322: It needs to be acknowledged that less gene expression for collagens is unexpected and surprising, since it is well established (Stager et al [17]) that the inferior oblique muscle contains more collagen when it is primary IOAA. Also, the increased collagen in primary IOAA makes the muscle stiffer, not more elastic (Stager et al [17]). This issue cannot be resolved without a direct comparison of gene expression with a baseline normal inferior oblique muscle (as correctly stated on line 323).

Figure 4 legend: As mentioned above, please explain how “myelination in the PNS” differs from “myelination”.

·

Basic reporting

1. The plagiarism rate was notably high at 36%. I will attach the plagiarism file. This percentage seems excessive, as the expected rate should be 17%.
2. Abstract formatting: Consider using standard structured subheadings (Background, Methods, Results, Conclusion) per journal format.
3. Language polishing: Some minor grammar/style issues remain:
• Line 224: Replace "invovled" with "involved" (Fig. 3 caption)
• Sentences like: These results provide insights into the increased presence of multiply innervated muscle fibers...", could be simplified or split for clarity.

Experimental design

4. Define abbreviations at first use in the abstract (e.g., EOM, DEG, GSEA, PPI).
5. Control limitations:
• As the authors noted, the absence of a healthy control group is a limitation. It may be beneficial to include this explicitly in the Discussion (Limitations) paragraph.
6. Patient demographics: Slightly expand Table 1 to include ranges for age and IOOA severity to better visualize the spread.
7

Validity of the findings

7. Clarify diagnosis criteria: Clearly state if one or more pediatric ophthalmologists confirmed IOOA diagnosis and grading
8. Interpretation of fold change: Provide additional context regarding the biological significance of the log2 fold change thresholds
9. Clarify the results by explicitly mentioning key findings from each supplementary table (S1, S2)
10. Include figure legends within the main text, placing them after the first mention of the figure rather than placing them separately later
11. Tables: improve spacing and alignment (Table 3 especially)
12. Supplement: Reference all supplementary materials in-text

Additional comments

Finally
This study provides a valuable and methodologically robust transcriptomic analysis comparing primary and secondary IOOA using RNA sequencing (RNA-seq). Although the sample size is modest, it is suitable for exploratory molecular profiling. The authors have shown a commitment to ethical standards and technical rigor. I recommend some minor edits for clarity in language and improvements in data visualization. I support the acceptance of the study after these revisions are made.